# Chemical Modification of Poly(1-Trimethylsylil-1-Propyne) for the Creation of Highly Efficient CO_2_-Selective Membrane Materials

**DOI:** 10.3390/ma12172763

**Published:** 2019-08-28

**Authors:** Viktoriya Polevaya, Viktoriya Geiger, Galina Bondarenko, Sergey Shishatskiy, Valeriy Khotimskiy

**Affiliations:** 1A.V. Topchiev Institute of Petrochemical Synthesis RAS, Leninsky pr., 29, 119991 Moscow, Russia; 2Institute of Polymer Research, Helmholtz-Zentrum Geesthacht, Max-Planck-Strasse 1, 21502 Geesthacht, Germany

**Keywords:** 1,2-disubstituted polyacetylenes, CO_2_ separation, ionic liquids, polymeric membranes, gas separation

## Abstract

The work is devoted to the chemical modification of a polymer that is promising for the creation of gas separation membranes, aimed at increasing the selectivity with respect to CO_2_. The introduction of ionic liquids into the structure of poly(1-trimethylsilyl-1-propyne) is realized by a two-step process: bromination of the initial polymer with *N*-bromosuccinimide and subsequent addition of tertiary amine (*N*-butylimidazole) to it. Depending on the process conditions, the method allows polymers with different contents of the ionic liquid to be obtained. The obtained polymers show good film-forming properties and thermal stability. Depending on the content of the ionic liquid in the polymer matrix, the resistance to aliphatic alicyclic to the majority of halogenated, as well as aromatic hydrocarbons, increases. With an increase of the ionic liquid content in the polymer, the ideal selectivities of CO_2_/N_2_ and CO_2_/CH_4_ gas pairs increases while maintaining a high level of permeability.

## 1. Introduction

The problem of carbon dioxide separation from various industrial products is currently one of the most important global problems. The issue of climate change is related to the emission of CO_2_ into the atmosphere, which is mainly a result of rapidly growing energy consumption and production of energy from fossil fuels [1].

The main directions for reducing carbon dioxide emissions are flue gas streams treatment (CO_2_/N_2_) and the purification of natural energy sources (natural gas, biogas; CO_2_/CH_4_) to increase the heat of combustion and prevent corrosion of pipelines [2]. Thus, efficient separation of CO_2_ from light gases, such as CH_4_ and N_2_, is a key technical, economic and environmental problem.

Various methods for capturing CO_2_ from different gas mixtures of industrial origin, such as chemical [3,4] and physical absorption [5], adsorption [6,7] or cryogenic distillation [8], are being developed and investigated. In addition to the aforementioned methods, in recent years, membrane technology for the separation of CO_2_ from gas mixtures of various compositions has been intensively developed [9,10]. These processes are characterized by high efficiency of gas separation, low energy consumption, simplicity of hardware design and thus relatively low capital and operation costs.

The main characteristics of the polymer that determine its successful use in real-life processes of CO_2_ membrane separation are a combination of a high level of gas permeability and CO_2_ selectivity with stability toward the gas mixture components and the mechanical and thermal stability of the membrane material under operating conditions. However, currently used polymer materials do not fully meet these requirements due to either low permeability or stability. In this regard, the chemical modification of polymers is of particular interest, allowing one not only to improve the performance of already known materials, but also to create completely new membrane materials that have the desired characteristics.

The transport of gas molecules through a non-porous polymeric layer can be described in the framework of the solution–diffusion mass transfer mechanism. In accordance with this mechanism, the permeability of the membrane (*P*) can be represented as the result of two processes, solubility (*S*) and diffusion of the penetrant (*D*) [11]:*P* = *D* × *S*(1)

The efficiency of the separation is expressed by selectivity, $\alpha_{A/B}$, determined as the ratio of the permeability coefficients of two gases, A and B:(2)$$\alpha_{A/B}=\frac{P_{A}}{P_{B}}=\frac{D_{A}}{D_{B}}\times \frac{S_{A}}{S_{B}}$$

The effective use of membranes can be based either on the size and shape differences of the separated molecules or on specific interactions of the molecules with the membrane material. On this basis, one of the promising types of chemical modification of polymers is the introduction of various functional substituents into the polymer structure to improve the solubility of the required gas in the polymer matrix.

One of the options for such substituents would be the use of room temperature ionic liquids, which are successfully used as functional materials for various applications, including gas separation processes. In accordance with the generally accepted definition, room temperature ionic liquids are salts consisting of ions (organic cations and organic or inorganic anions) that have melting points below 100 °C. The most significant feature of ionic liquids is the highly selective dissolution of carbon dioxide compared to other gases. Thus, the introduction of ionic liquids into the polymer matrix generally leads to an increase in the selectivity of CO_2_ separation from various gas mixtures. However, in this case, a significant decrease in the permeability coefficients of individual gases is usually observed [12,13,14] due to the general trend among functionalized polymers in which an increase in selectivity of separation is accompanied by a drop in permeability and vice versa [15]. Therefore, the modification of highly permeable materials such as e.g., 1,2-disubstituted polyacetylene poly(1-trimethylsilylpropyne-1) (PTMSP), which one of the fastest in terms of gas permeability coefficients among the currently studied polymers, can be particularly promising, as a high level of initial permeability will allow some of it to be sacrificed to achieve the required selectivity values.

Despite the fact that obtaining functionalized polyacetylenes of different structures as new materials and investigating their properties for membrane separation are important tasks, this class of polymers is still not extensively studied. The information available in the literature on the gas transport properties of substituted polyacetylenes containing ionic liquids in their structure is mainly limited to polymers based on phenyl- and diphenylacetylene derivatives containing imidazole salts as a part of the aromatic substituents [16,17]. In particular, as noted in the work [16], the surface modification of membranes from poly(diphenylacetylenes) containing a bromoethoxy- group with 1-methylimidazole resulted in the CO_2_/N_2_ selectivity increase from 16 to 44. An example of the introduction of imidazole fragments into the polymer matrix of the PTMSP carbon analogue, poly(4-methyl-2-pentyne) (PMP), by heterogeneous modification is known. Quaternization of *N*-butylimidazole with bromine-containing PMP resulted in a CO_2_/N_2_ selectivity increase and, in comparison to the initial polymer, resistance to cycloaliphatic hydrocarbons was also increased [17]. At the same time, in all the aforementioned cases, polymers with a relatively low initial gas permeability were studied. Therefore, along with the increased selectivity, the functionalized polymers exhibited a reduced level of permeability, insufficient for practical use. Thus, there is no information in the literature on the modification of 1,2-disubstituted polyacetylenes leading to an optimal combination of gas transport parameters. However, the reported tendencies of transport properties change due to modification, open the possibility of targeted change of gas transport properties by introduction of CO_2_-specific groups into the chemical structure of the polymer.

In this work, ionic liquids based on *N*-butylimidazole were used as functional groups for polyacetylenes modification, since imidazole ionic liquids with an alkyl side chain of similar length exhibit high CO_2_ solubility. The PTMSP with ionic liquids as the side substituents cannot be obtained directly from the metathesis polymerization reaction, as the polar functional groups deactivate the active polymerization sites. Therefore, the introduction of ionic liquids into the polymer structure was carried out by a two-step process: bromination of the starting polymer with *N*-bromosuccinimide and subsequent quaternization reaction of *N*-butylimidazole with the resulting bromine-containing polymer (Figure 1).

## 2. Materials and Methods

### 2.1. Materials

1-(Trimethylsilyl)-1-propyne) (TMSP; 99.8%, AO Yarsintez, Yaroslavl, Russia) was distilled twice from calcium hydride (CaH_2_) in a flow of high-purity argon at atmospheric pressure and was stored in the same atmosphere.

Tantalum pentachloride (TaCl_5_; anhydrous, powder, 99.999%; Sigma-Aldrich, Steinheim, Germany) and triisobutylaluminum (TIBA; 25 wt.% (1.0 M) solution in toluene; Sigma-Aldrich, Steinheim, Germany) were used without additional purification. For dosed addition to the reaction mixture, powdered TaCl_5_ was preliminarily packed in glass ampules, which were then sealed under high-purity argon. The sealed ampules with the catalyst were stored at a temperature not higher than 4 °C.

Toluene (for HPLC, 99.9%; Acros Organics, Geel, Belgium) was dried and distilled over CaH_2_ directly before polymerization.

*N*-Butylimidazole (98%, Acros Organics) was distilled over CaH_2_ under vacuum.

Tetrahydrofuran (THF; HPLC grade, Chimmed, Moscow, Russia) was passed through an Al_2_O_3_ column, dried over KOH (in pellets), heated at reflux and distilled over CaH_2_ in high purity argon atmosphere. After that, THF, benzophenone and sodium (small flakes) were heated at reflux until the ketyl was formed, which was then distilled in high purity argon atmosphere.

### 2.2. Synthesis of PTMSP on TaCl_5_–TIBA

The glass polymerization reactor equipped with a magnetic stirrer, a thermometer, and an inert gas (argon) inlet was charged with the powdered TaCl_5_ catalyst by opening the glass ampule, and the amounts of the cocatalyst (TIBA alkylating additive), solvent (toluene) and monomer were calculated from the TaCl_5_ weight to ensure the following reaction conditions: TaCl_5_/TIBA molar ratio 0.3, monomer/cocatalyst molar ratio 50, monomer concentration in the solution 1 M [18]. After TaCl_5_ loading, toluene was added, the temperature in the reactor was elevated to 80 °C, and the mixture was stirred until the catalyst dissolved completely, after which the solution was cooled to room temperature. After adding the required amount of TIBA from the Schlenk vessel, the mixture was stirred for 1 h at room temperature to form a catalytic complex. TMSP monomer was added with vigorous stirring at 2 °C. The polymerization was performed for 24 h. Then, the rubber-like polymerization product was unloaded, shredded to fine particles, and mixed with a 20% solution of methanol in toluene to decompose the catalyst. After that, toluene was added in an amount required to obtain a 1.5% PTMSP solution, and the mixture was stirred until the polymer was completely dissolved. The resulting solution was filtered through a gauze filter, and the product was precipitated into a six-fold excess of methanol. Then, the polymer was filtered off, washed with methanol, and dried for 8 h at ambient conditions, after which the dissolution, precipitation, and drying of the polymer were repeated. The resulting polymer was dried in a vacuum for 8 h, and the yield was determined.

The ratio of the cis/trans units in PTMSP synthesized on the TaCl_5_/TIBA catalytic system was 35/65, Mw = 3 × 10^6^, Mw/Mn = 1.9, the intrinsic viscosity [η] (25 °C, CCl_4_) = 6.5 dL/g.

### 2.3. Synthesis of Brominated PTMSP

Brominated PTMSP was prepared according to the procedure described in [19]. It was found that for quaternization it is optimal to use PTMSP containing 60 mol% of brominated monomeric units, as the polymers with such bromination degree show good mechanical properties, thermal stability, high permeability and already have a relatively high selectivity for the separation of CO_2_ from gas mixtures containing N_2_ and CH_4_.

### 2.4. Quaternization of N-Butylimidazole by the Brominated PTMSP

The quaternization reaction was carried out in a temperature-controlled oil-jacketed reactor in a high purity argon stream. A solution of the brominated polymer (approx. 5 wt.% PTMSP in THF) was charged into the reactor and heated to 55 °C, the temperature was kept constant throughout the experiment. After that, *N*-butylimidazole was added at varying molar ratios (from 1 to 10 moles per mole of polymer units). The reaction time was 72 h. The reaction mixture was stirred with a magnetic stirrer for the duration of the reaction. By the end of the reaction, the reaction mixture was poured dropwise into methanol, filtered through a glass Schott filter, and the polymer was washed several times with methanol and water to remove unreacted *N*-butylimidazole and, finally, dried under vacuum to constant weight.

### 2.5. Physico-Chemical Characterization

The values of the number-average molecular (Mn) and weight-average molecular (Mw) weights of the synthesized PTMSP were determined by gel permeation chromatography with polymer solutions in toluene. The measurements were carried out on a Waters chromatograph (column—Chrompack Microgel-5 Mix R-401; solvent—toluene; temperature—25 °C; flow velocity—1 mL/min; standard—polystyrene). The calibration plot was obtained for an Mw range of 4000–2,000,000.

The intrinsic viscosity of polymer solutions in CCl_4_ were determined with an Ostwald–Ubbelohde viscometer at 25 °C. The amount of nitrogen in the polymer was determined by the element–organic analysis using a Perkin Elmer 2400 Series II Elemental Analyzer (PerkinElmer Inc, Waltham, MA, USA). The number of attached *N*-butylimidazole units was calculated from the nitrogen content.

Recalculation of the mass content of nitrogen in the molar content of quaternized units was carried out according to the formula:*Φ = xM3/(28 – x(M1 – M2 – M3))*(3) where Φ is the molar fraction of *N*-containing units, x is the mass fraction of nitrogen in the sample (according to the results of elemental analysis), 28 is the relative atomic mass of nitrogen, M1, M2 and M3 are the molecular (molar) masses of the initial, brominated and quaternized elementary unit of PTMSP, respectively.

The position of bromine in the PTMSP unit and confirmation of butylimidazole unit presence in the polymer was determined by IR spectroscopy in the 500–4000 cm^–1^ range with a IFS-Burker-113-V spectrometer (Germany), 30 scans, resolution 2 cm^−1^. Samples were prepared in the form of films cast from the polymer solution in THF.

The quantitative ratio of the cis- and trans- structures in PTMSP was calculated from the ^13^C NMR spectra using the Winnmr1d program (Bruker, Billerica, MA, US) suitable for calculation of incompletely resolved spectra. The ratio was calculated from the relative intensities of the peaks in the doublets. The peaks were assigned in [18] by combining the NMR and IR data with theoretical analysis of the normal modes of polymers with different microstructures.

Thermal was carried out on “Mettler Toledo DSC-1” (Mettler Toledo GmbH, Greifensee, Switzerland) instrument in the range 20–1000 °C. The measurements were done in air and argon atmospheres at 10 °C/min heating rate for 5 to 40 mg samples placed in crucibles of aluminium oxide with a volume of 70 μL. The accuracy of the measurements to determine the temperature was ±0.3 °C, and to determine the mass it was ±0.1 μg.

Differential scanning calorimetry (DSC) was carried out with a differential scanning calorimeter Mettler Toledo DSC823e (Mettler Toledo GmbH, Greifensee,, Switzerland) in the range 20–350 °C. The samples with weight from 5 to 20 mg were placed in 40 μL crucibles of aluminium oxide and closed with a perforated cover. The perforation of the crucible cover provided a free exchange with the furnace atmosphere ensuring measurements at constant pressure. Heating of the samples was carried out at 20 °C/min rate in an argon atmosphere with a flow rate of 70 mL/min. The measurement results were processed using the STARe service program supplied with the DSC equipment. The temperature was determined with ±0.1 °C accuracy.

To evaluate the polymer solubility, the polymer sample (100 mg) was kept in a suitable solvent (25 mL) for 48 h at 25 °C, and then heated to 60 °C for 6 h. Then, the solution was filtered and the filtrate was precipitated into methanol. The solubility of the polymer was evaluated from the results of weighing.

X-ray diffractograms of the polymer samples were obtained using an X-ray source with a rotating copper anode (Rigaku Rotaflex RU-200, Rigaku, Tokyo, Japan) equipped with a horizontal wide-angle goniometer (Rigaku D/Max-RC). θ–2θ scanning was carried out according to the Bragg–Brentano scheme in the reflection geometry, with the film sample fixed on an aluminium frame-holder. The measurement range of the diffraction angles was 2.5–50° to 2θ, the measurement was carried out in the continuous scan mode at a rate of 4 °/min and a step of 0.04°. A scintillation counter played the role of a detector of diffracted X-ray radiation, the incident radiation on it was monochromatized with the aid of a secondary (i.e., standing on a beam reflected from the sample) focusing monochromator, a curved monocrystal graphite. The wavelength of monochromatic radiation was 1.542 Å. The position and half-width of the X-ray maxima were determined by using the Fityk program [20], the peaks being approximated by the Gaussian (normal distribution) function. The characteristic interchain distance in the polymer was determined from the angular position of the corresponding maximum by using the Wolf–Bragg formula.

Polymeric isotropic membranes with film thicknesses in the range 25–50 μm were obtained by casting from 1 to 5 wt.% solution in THF on a levelled cellophane support and covered with a Petri dish for slow evaporation of the solvent. The films were air dried for 7 days and then dried under vacuum for 48 h.

The permeability, diffusion and solubility coeffcients of polymers were determined for individual gases at 30 °C by using an experimental facility, operating on the principle of “constant volume/variable pressure”, designed to measure the gas transport properties of flat membranes with known thickness [21,22,23]. This method is based on the flow rate measurement of gas passing through the membrane, which consists of determining the pressure change in the calibrated volume per unit time. The permeability coefficients of the membranes (*P*) were determined by the expression:(4)$$P=D\times S=\frac{V_{p}l}{ART\Delta t}\ln\frac{p_{f}-p_{p1}}{p_{f}-p_{p2}}$$ where *Vp* is the volume of the permeate, *l* is the membrane thickness (25–50 μm), *A* is the membrane area, *R* is the universal gas constant, *p_f_* is the feed pressure (1 bar for all gases in the time interval Δ*t*), *p_p1_* and *p_p2_* are the permeate pressures at time points 1 and 2, *Δt* is the time difference between two points (1 and 2) on the permeate pressure increase curve. The points 1 and 2 were taken from the linear part of the pressure increase curve, the time-lag determined from linear regression intersection with the time axis. All membranes were carefully degassed before gas transport experiments to avoid any influence of volatile compounds desorption, either gases or solvent, on experimental results. 

The value of the diffusion coefficient (*D*) was determined from the time-lag by using the Equation (4):(5)$$D=\frac{l^{2}}{6\theta }$$ where *l* is the membrane thickness, and *θ* is the time-lag.

The value of the solubility coefficients (*S*) was calculated from the equation:(6)$$S=\frac{P}{D}$$

The ideal selectivity of the polymer films was calculated from the ratio of the permeability coefficients of the individual gases A and B:(7)αAB=PAPB where *α(A*/*B)* is the ideal selectivity, *P_A_* and *P_B_* are the permeability coefficients of the two gases, *A* and *B*, with the gas *A* having higher permeability coefficient.

## 3. Results and Discussions

### 3.1. Synthesis and Properties of PTMSP Containing Butylimidazole Bromide

It was found that PTMSP containing 60 mol% of brominated monomeric units is the best choice for further quaternization, because the polymers with such a bromine content show good mechanical properties, thermal stability, high permeability and already have a relatively high selectivity for the separation of CO_2_ from mixtures with N_2_ and CH_4_.

The conditions and results for the reaction of bromine-containing poly(1-trimethylsilyl-1-propyne) and N-butylimidazole are shown in the Table 1.

As follows from the Table 1, it is possible to obtain PTMSP samples with varying contents of butylimidazole salt in a polymer matrix by adjusting the reagent ratios. The maximum degree of quaternization reached 20 mol% under the chosen conditions.

The IR spectra of the original PTMSP, bromine-containing and quaternized polymer in comparison with the spectrum of *N*-butylimidazole are shown in Figure 2.

Two new absorption bands of 580 cm^–1^ (ν_C–Br_) and 1221 cm^–1^ (δ_SiCH2_) appear in the brominated PTMSP (Figure 1 and Figure 2), indicating the formation of the covalent bond -Si(CH_3_)_2_CH_2_–Br in the polymer. [20] The presence of an imidazole ring in the polymer structure can be judged from the appearance of new bands in the 1680–1720 cm^–1^ region and new weak bands in the 1250–1000 cm^–1^ region. In this case, a strong change in the spectrum in the region of valence vibrations of the C–H and C=C bonds of the aromatic ring (3110 and 1500 cm^–1^) indicates a significant rearrangement of the butylimidazole in the π-electron structure. Similar changes in the IR spectrum of *N*-butylimidazole were observed when imidazole chloride appeared, as described in the work [24].

Changes in the spectrum of the quaternized polymer are also observed for bands characterizing the bonds at the C=C–Si(CH_3_)_2_CH_2_–Br site: the band is broadened and shifted from the double bond of 1536 cm^–1^, the band shifts from 1221 cm^–1^ (δ_SiCH2_) to 1217 cm^–1^, and the 680 cm^–1^ band, related to the bond stretching vibrations =C–Si [25], shifts to the region of long waves and strongly increases in intensity.

Such changes in the spectrum can be explained by a change in the polarization of the C=C and Si–C bonds due to the formation of an ion pair: the imidazole cation and the Br^–^ anion bound to the polymer.

Data on the solubility of functionalized polymers in organic solvents are presented in Table 2.

Samples with up to 5 mol% butylimidazole salt content in the polymer matrix are soluble in polar solvents such as THF and CHCl_3_, and when the content of the quaternized units increases, they also become resistant to the action of less polar solvents—aliphatic alicyclic C_5_–C_12_, cycloalkanes (cyclohexane), halogenated solvents (CCl_4_) and also aromatic hydrocarbons. When the content of the quaternized units is increased to 20 mol%, the polymers become completely insoluble in organic solvents.

Differences in solubility can be associated to the packing morphology of the studied polymers. Figure 3 shows the diffraction curves of PTMSP (a), a polymer containing 60 mol% bromine (b) and polymers obtained on its basis with different contents of imidazole salt (c,d).

In the diffractograms of all samples, a basic diffusion maximum corresponding to the interchain periodicity (2θ~10°) is present [26]. The values of the half-width (Δ_1/2_, °) of the main reflection (Table 3) indicate a small size of the coherent scattering regions, but still much larger than for truly amorphous polymers, for which the half-width of the reflection is usually 5–8° [27]. 

The X-ray patterns of the bromine-containing polymer (Figure 3b) and butylimidazole salt-enriched polymers (Figure 3c,d) demonstrate not only the main reflection, but also additional diffuse maxima at 2θ~20° and 25°. Some increase in the interplanar distance (*d*, Å) in polymers containing bromine and imidazole salts in comparison with the initial polymer appears to be the result of the introduction of bulky side substituents into the macromolecules.

The relatively high values of the coherent scattering regions and the presence of extra reflections, in addition to the most intensive one, indicate that the polymers have a two-phase nature of the supramolecular structure, which consists of less regular regions and regions with an elevated level of regularity. This morphology is characteristic for glassy polymers of the acetylene series, such as PTMSP, PMP and PTMGP [28]. The presence of additional reflections, the intensity of which increases with the content of the butylimidazole salt in the polymer matrix, may indicate an increase in the regularity of the structure of the functionalized polymers depending on the quantitative content of the functional groups, which corresponds well with polymer solubility (Table 2). Polymers with the most intense additional reflections from the diffraction curves have the most regular structure, which is probably the reason for the loss of polymer solubility.

As follows from the TGA results, the temperature of the beginning of weight loss for the polymer decreases with the increasing content of quaternized units of butylimidazole in the polymer structure (Figure 4).

In an inert gas atmosphere for PTMSP containing 5 mol% quaternized units, the curve shows the onset of decomposition (3% mass loss, dashed line on the Figure 4) at 215 °C. The mass loss of the polymer containing 20 mol% butylimidazole salts begins at 202 °C.

To investigate the relaxation properties of polymers, the DSC method was used. Analysis of the DSC curves showed that polymers, as well as the initial bromine-containing PTMSP, do not show signs of a glass transition or fluidity in the temperature range 0–280 °C. Probably in this case all the relaxation transitions associated with a glass transition and fluidity are above the decomposition temperature.

### 3.2. Gas Transport Characteristics of PTMSP Containing Butylimidazole Salts

In Table 4, the results of measurements of transport characteristics for individual gases are presented for quaternized samples of different compositions. It should be noted that the film-forming properties deteriorate when the content of imidazole in the polymer is more than 5%, which makes it difficult to use them as thin films.

The quaternization reaction of butylimidazole with the bromine-containing PTMSP resulted in a decrease in the gas permeability and in a significant increase in the ideal selectivity of CO_2_ over other gases of interest. The decrease in CO_2_ gas permeability occurred mainly because of the decrease in the diffusion coefficient, which, apparently, was caused by the increased intermolecular interaction due to the presence of butylimidazole salts in the polymer structure. At the same time, the solubility coefficient of CO_2_ decreased by 35%, while other gases under study demonstrate up to 82% of solubility coefficient loss.

Table 5 shows that the ideal selectivities for CO_2_/N_2_ and CO_2_/CH_4_ gas pairs in comparison to O_2_/N_2_ gas pair, where no specific interaction caused by salt introduction into the polymer structure was expected, for all samples is higher than that of the bromine-containing PTMSP.

When the optimal number of butylimidazole bromide units is present in the polymer matrix, the CO_2_/N_2_ and CO_2_/CH_4_ permeability selectivities increase approximately 2-fold compared to the initial brominated polymer. The diffusivity selectivities of D_CO2_/D_N2_ and D_CH4_/D_N2_ for the polymers containing imidazole salts decreased almost twice compared to the bromine-containing PTMSP, while the same for O_2_/N_2_ gas pair increased by 30%. This is a very interesting effect of the imidazole salts presence in the polymer matrix which cause significant reduction of the CO_2_ diffusion coefficient, the gas with the smallest kinetic diameter. At the same time, the S_CO2_/S_N2_ solubility selectivity for quaternized polymers increases from 7.1 to 26.4 in comparison to bromine-containing PTMSP, and the solubility selectivity of S_CH4_/S_N2_ increases from 3.3 to 8.8. Thus, the significant increase in CO_2_/N_2_ and CO_2_/CH_4_ selectivity is based on the increased selectivity of dissolution, which is related to the active interaction of butylimidazole salt with the CO_2_ molecule. At the same time, it is necessary to note that CO_2_ at 30 °C is reversibly sorbed in the synthesized polymers in contrast to e.g., quaternary ammonium compounds studied in [29] which can be judged from the time-lag and the corresponding diffusion coefficients (Figure 5). This is confirmed by the analysis of the “time-lag” curves for CO_2_, CH_4_ and N_2_. The “time-lag” curves of CO_2_ accumulation in the permeate volume of the experimental facility obtained one after another with a short, 2 min evacuation time between them, show exactly the same behaviour as observed for other two gases presented on Figure 5. The time-lag deviation is very minor which results in just 1.2% error of CO_2_ diffusion coefficient fluctuation while the same value for CH_4_ is 6.3%. The experimental facility used for gas transport experiments has just 35 ms of systematic error of the time-lag determination and in the current study showed ±5% error in permeability coefficient detemination.

In order to compare properties of studied polymers with those of polymers reported in the literature, the selectivity–permeability relationship for the separation of the CO_2_/N_2_ gas pair for different polymers [15,30], is shown on Figure 6.

Analysis of the polymer position with respect to the upper bound of the Robeson diagram, determined in 2008 [15], makes it possible to evaluate the effectiveness of its ideal gas separation properties, that is, the closer the polymer is to the boundary, the more effective it is for the selected separation process. It should be noted that PTMSP, containing 5 mol% butylimidazole bromide units, is located relatively close to the upper boundary and demonstrates improved gas separation characteristics compared to the initial and brominated polymers. Therefore, it can be considered as a promising material for use in the separation of CO_2_-containing gas mixtures.

## 4. Conclusions

A new polymer material, combining increased CO_2_ selectivity with high gas permeability, based on a silicon-containing glassy polymer of 1,2-disubstituted acetylenes, poly(1-trimethylsilyl-1-propyne), which was modified by introducing an ionic liquid, butylimidazole bromide as a side substituent, has been developed in this work.

The obtained polymer shows good film-forming properties up to the butylimidazole content of 5% and good thermal–oxidative stability in combination with attractive gas transport characteristics. It is shown that the permeability selectivity of CO_2_/N_2_ and CO_2_/CH_4_ gas pairs increases with the increase of the ionic liquid content in the polymer, while maintaining a high level of permeability. In addition, depending on the salt content in the polymer matrix, its resistance to aliphatic alicyclic, halogenated, as well as aromatic hydrocarbons increases.

Thus, the method of modification of PTMSP by adding functional groups is a promising way of creating highly effective membrane materials with increased selectivity for CO_2_ separation from industrial gas mixtures of various compositions.

## Figures and Tables

**Figure 1 materials-12-02763-f001:**
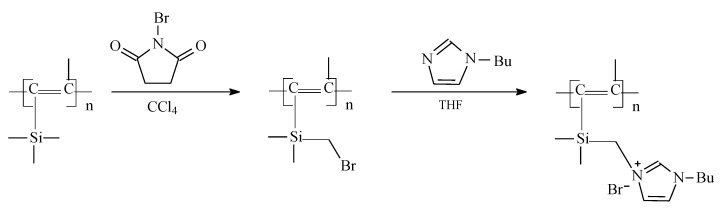
Modification of PTMSP by the butylimidazole salt.

**Figure 2 materials-12-02763-f002:**
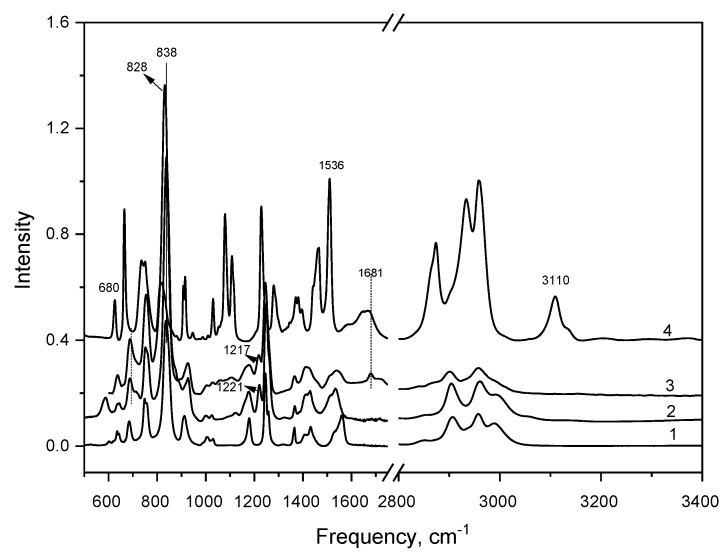
The IR spectra of PTMSP (1 PTMSP, containing 60 mol% Br (2), PTMSP, containing 5 mol% butylimidazole salt (3), *N*-butylimidazole (4).

**Figure 3 materials-12-02763-f003:**
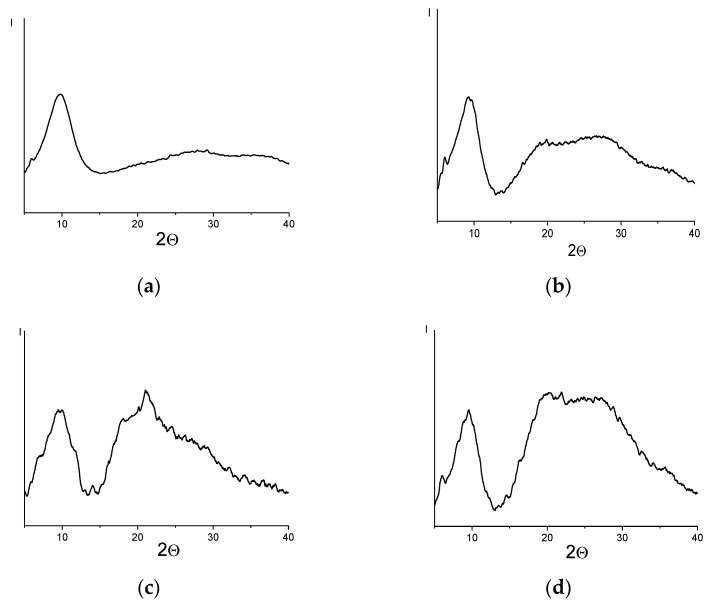
The diffraction curves of PTMSP (**a**), Br-containing PTMSP (**b**), polymer containing 5 mol% (**c**) and 20 mol% (**d**) butylimidazole salt.

**Figure 4 materials-12-02763-f004:**
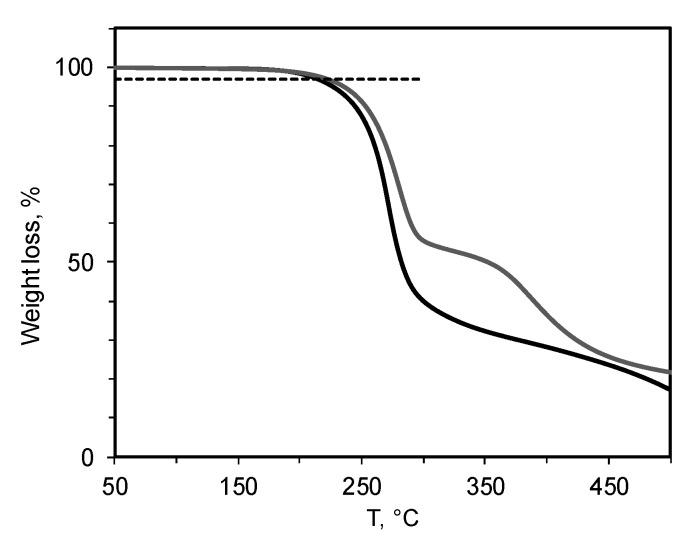
PTMSP with different contents of butylimidazole salt: 5 mol% (1) and 20 mol% (2). The dashed line represents the level of 3% of mass loss.

**Figure 5 materials-12-02763-f005:**
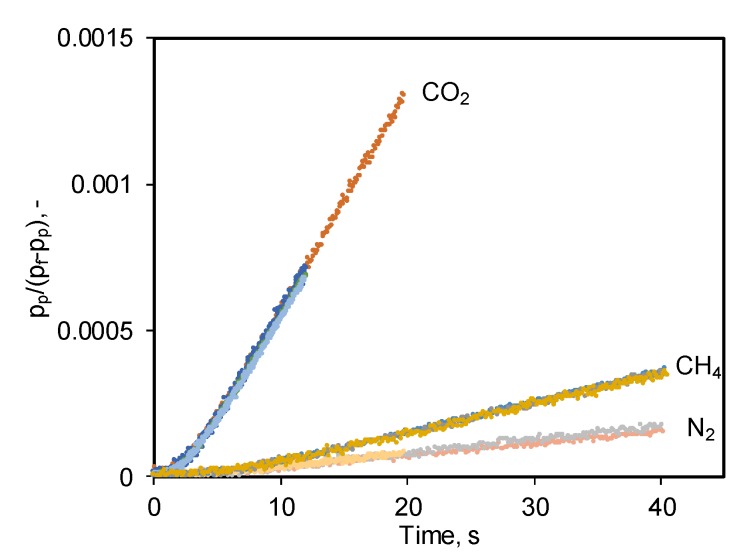
“Time-lag” curves for CO_2_, CH_4_ and N_2_ obtained for the sample of PTMSP, containing 5 mol% butylimidazole bromide units. The vertical axis represents the value of permeate pressure normalized by the driving force (p_f_−p_p_) for each acquired data point.

**Figure 6 materials-12-02763-f006:**
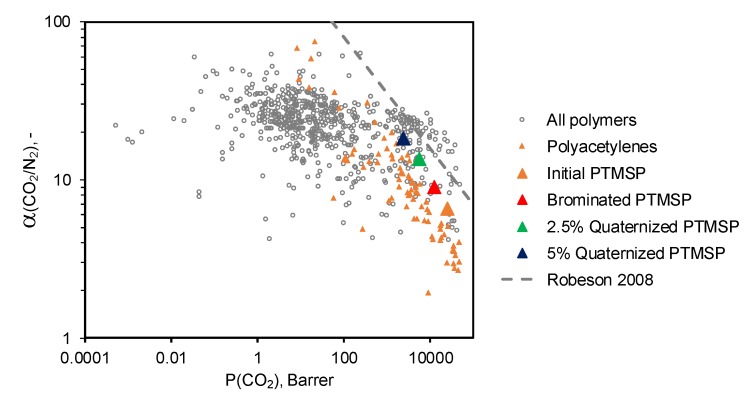
Position of studied polymers on the Robeson plot with 2008 upper bound for CO_2_/N_2_ separation. The gas transport data for polymers reported in the literature are taken from the Polymer Gas Separation Membrane Database of the Membrane society of Australasia [30].

**Table 1 materials-12-02763-t001:** Conditions and results of PTMSP quaternization (T of reaction = 55 °C, solvent—THF, concentration of bromine-containing PTMSP in solution—5 wt%, reaction time—72 h).

[Br]/[*N*-Butylimidazole] [mol/mol]	Content of N in the Polymer [wt%]	Content of Quaternized Units in the Polymer [mol%]
1:2	0.4	2.5
1:5	0.8	5.0
1:8	1.7	10.0
1:10	3.1	20.0

**Table 2 materials-12-02763-t002:** Solubility of PTMSP with different contents of butylimidazole units towards organic solvents ^1^.

Content of Quaternized Units in the Polymer [mol%]	THF	CHCl_3_	Toluene, Benzene	CCl_4_	Cyclohexane	C_5_–C_12_ ^3^
0 ^2^	+	+	+	+	+	–
2.5	+	+	+	–	–	–
5	+	+	–	–	–	–
10	±	±	–	–	–	–
20	–	–	–	–	–	–

^1^ “+”—soluble; “±”—partially soluble;“−“—insoluble. ^2^ the initial sample contains 60 mol% Br. ^3^ aliphatic alicyclic hydrocarbons C_5_–C_12_.

**Table 3 materials-12-02763-t003:** Radiographic characteristics of PTMSP samples with different contents of imidazole salt.

Content of Quaternized Units in the Polymer [mol%]	2*θ* [°], Basic Reflex	Δ_1/2_ [°]	Interplanar Distance *d* [Å]
0 ^1^	9.8	3.2	9.0
0 ^2^	9.3	3.0	9.5
5	9.6	3.7	9.3
20	9.3	3.4	9.5

^1^ initial PTMSP. ^2^ PTMSP containing 60 mol% bromine.

**Table 4 materials-12-02763-t004:** Gas permeability, solubility and diffusion coefficients of membranes containing butylimidazole salt in the polymer matrix.

Content of Quaternized Units in the Polymer [mol%]	P [Barrer] ^1^				D × 10^7^ [cm^2^/s]				S × 10^3^ [cm^3^(STP)/ (cm^3^ × cmHg)]			
	O_2_	N_2_	CO_2_	CH_4_	O_2_	N_2_	CO_2_	CH_4_	O_2_	N_2_	CO_2_	CH_4_
0 ^2^	2640	1400	12,500	2880	55	35	44	34	48	40	284	85
2.5	1200	406	5480	836	50	27	28	19	24	15	197	44
5	429	133	2410	336	39	19	13	16	11	7	185	21

^1^ 1 Barrer = 10^–10^ cm^3^(STP) × cm × cm^–2^ × s^–1^ × cmHg^–1^. ^2^ PTMSP containing 60 mol% bromine.

**Table 5 materials-12-02763-t005:** Estimated selectivity of polymers containing butylimidazole salts.

Content of Quaternized Units in the Polymer [mol%]	O_2_/N_2_			CO_2_/N_2_			CO_2_/CH_4_		
	α_P_	α_D_	α_S_	α_P_	α_D_	α_S_	α_P_	α_D_	α_S_
0 ^1^	1.9	1.6	1.2	8.8	1.3	7.1	4.4	1.3	3.3
2.5	3.0	1.9	1.6	13.5	1.0	13.1	6.6	1.5	4.5
5	3.2	2.1	1.6	18.1	0.7	26.4	7.2	0.8	8.8

^1^ PTMSP containing 60 mol% bromine.

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
