# Peer review of "Chemical Modification of Poly(1-Trimethylsylil-1-Propyne) for the Creation of Highly Efficient CO_2_-Selective Membrane Materials"

_materials, 2019, doi:10.3390/ma12172763_

Round 1

Reviewer 1 Report

This study reports the preparation of a new polymer material combining increased CO2 selectivity with high gas permeability.The authors discovered that chemical modification of poly(1-trimethylsylil-1-2 propyne) membranes material by introducing an ionic liquid (butylimidazole bromide) it leads to high selectivity towards CO2. Overall, I found the results are very interesting and convincing and insights are correct.  I support its publication. 

My suggests are placed in the text of the paper:

page 9, lines 296-299: I propose show the curves page 11, lines 349-352: In my opinion that sentences not is a conclusion from the results obtained. This is a statement of what has been done at work. I propose of the change that sentences

Author Response

My suggests are placed in the text of the paper:

Answer: Unfortunately the link on the MDPI site which should lead to the text of the current manuscript leads to the work of other authors (materials-572613-review.pdf) Influence of surface micro-patterning and hydrogel coating on colloidal silica fouling of polyamide thin-film composite membranes  by Ibrahim M.A. ElSherbiny , Ahmed S.G. Khalil  and Mathias Ulbricht

page 9, lines 296-299: I propose show the curves page 11, lines 349-352:

Answer: Corrected

In my opinion that sentences not is a conclusion from the results obtained. This is a statement of what has been done at work. I propose of the change that sentences

Answer: Thank you very much! We have changed the first sentence of the Conclusions.

Reviewer 2 Report

(1) The English writing needs to be improved throughout the manuscript. The current version is not easy to read.

(2) Line 111, Elaborate the polymer (PTMSP) synthesis, and provide the basic information for the obtained polymer product, for example, molecular weight (Mn, or Mw).

(3) Line 112, how the “content of trans-unit, 65%”, and intrinsic viscosity was determined?

(4) In Line 127, quaternization reaction was carried out at 60 deg C, while in Table 1, the reaction temp. is 55 deg C. Double check the reaction conditions.

(5) Figure 2, the title of x-axis should be wavenumber.

(6) Delete Figure 3, because it duplicates the information already shown in Figure 1.

(7) Redo the XRD experiment, the baselines of the diffraction patterns are not even. Typically, the XRD baseline of PTMSP should be flat, as shown in doi.org/10.1016/j.seppur.2015.11.032, doi.org/10.1016/j.micromeso.2007.06.044, or doi.org/10.1016/S0969-806X(00)00218-8.

(8) The baselines for the XRD patterns are so rough, so the peak half-width values might not be accurate.

(9) In Table 4, present the uncertainty of gas transport properties, including permeability, diffusivity, and solubility.

(10) The authors have noted that quaternization significantly reduces CO2 (with the smallest kinetic diameter) diffusion coefficient, rendering it even smaller than that of N2, O2, and CH4 with the larger kinetic diameter and making CO2/N2 (or CH4) diffusion selectivity is less than 1. Why, what is the possible reason? Is that possible resulting from the uncertainty in determining the diffusivity coefficient?

(11) It’s good to show the time-lag plots for the determination of diffusivity coefficients.

(12) In Figure 6, put citations for the data points of ‘all polymers’ and ‘polyacetylenes’

(13) Misuse the quotations marks, for example, "Acros Organics" in Line 119, the "reflection" geometry in Line 162.

Author Response

Reviewer 2

 (1) The English writing needs to be improved throughout the manuscript. The current version is not easy to read.

Answer: We have tried our best to correct the language.

(2) Line 111, Elaborate the polymer (PTMSP) synthesis, and provide the basic information for the obtained polymer product, for example, molecular weight (Mn, or Mw).

Answer. Done, the text is added and initial text revised:

2.1. Materials

1-(Trimethylsilyl)-1-propyne) (TMSP; 99.8%, AO Yarsintez, Russia) was distilled twice from calcium hydride (CaH2) in a flow of high-purity argon at atmospheric pressure and was stored in the same atmosphere.

Тantalum pentachloride (TaCl5; anhydrous, powder, 99.999%; Sigma-Aldrich) and triisobutylaluminum (TIBA; 25 WT. % (1.0M) solution in toluene; Sigma-Aldrich) were used without additional purification. For dosed addition to the reaction mixture, powdered TaCl5 was preliminarily packed in glass ampules, which were then sealed under high-purity argon. The sealed ampules with the catalyst were stored at a temperature not higher than 4°С.

Toluene (for HPLC, 99.9%; Acros Organics) was dried and distilled over CaH2 directly before polymerization.

2.2. Synthesis of PTMSP on TaCl5–TIBA

The glass polymerization reactor equipped with a magnetic stirrer, a thermometer, and an inert gas (argon) inlet was charged with the powdered TaCl5 catalyst by opening the glass ampule, and the amounts of the cocatalyst (TIBA alkylating additive), solvent (toluene) and monomer were calculated from the TaCl5 weight so as to ensure the following reaction conditions: TaCl5/ TIBA molar ratio 0.3, monomer/cocatalyst molar ratio 50, monomer concentration in the solution 1 M. After loading TaCl5, toluene was added, the temperature in the reactor was elevated to 80°С, and the mixture was stirred until the catalyst dissolved completely, after which the solution was cooled to room temperature. After adding the required amount of TIBA from the Schlenk vessel, the mixture was stirred for 1 h at room temperature to form a catalytic complex. TMSP monomer was added with vigorous stirring at 2°С. The polymerization was performed for 24 h. Then, the rubber-like polymerization product was unloaded, finely divided, and mixed with a 20% solution of methanol in toluene to decompose the catalyst. After that, toluene was added in an amount required to obtain a 1.5% PTMSP solution, and the mixture was stirred until the polymer dissolved completely. The resulting solution was filtered through a gauze filter, and the product was precipitated into a six fold excess of methanol. Then, the polymer was filtered off, washed with methanol, and dried in air for 8 h at room temperature, after which the dissolution, reprecipitation, and drying of the polymer were repeated. The resulting polymer was dried in a vacuum for 8 h, and its yield was determined.

The ratio of the cis/trans units in PTMSP synthesized on the TaCl5/TIBA catalytic system was 35/65, Mw=3×106, Mw/ Mn = 1.9, the intrinsic viscosity [η] (25 °C, CCl4) = 6.5 dL/g.

2.3. Physico-chemical characterization

The values of the number-average molecular weight (Mn) and weight-average molecular PTMSP weight (Mw) were measured by gel permeation chromatography with polymer solutions in toluene. The measurements were made on a Waters chromatograph (column _ Chrompack Microgel-5 Mix R-401; solvent _ toluene; temperature _ 25 °C; flow velocity_ 1 mL/min; standard _ polystyrene). The calibration plot was obtained for an Mw range of 4000–2,000,000 (4075, 32,600, 95,800, 400,340, 850,000, 1,447,000, and 2,000,000).

The intrinsic viscosity of the polymer solutions in CCl4 were determined with an Ostwald–Ubbelohde viscometer at 25 °C.

The quantitative ratio of the cis and trans structures in PTMSP was calculated from the 13С NMR spectra

using the Winnmr1d program (Bruker) suitable for calculation of incompletely resolved spectra. The

ratio was calculated from the relative intensities of the peaks in the doublets. The peaks were assigned in [33] by combining the NMR and IR data with theoretical analysis of the normal modes of polymers with different microstructures.

(3) Line 112, how the “content of trans-unit, 65%”, and intrinsic viscosity was determined?

Answer: The following explanation is added:

The quantitative ratio of the cis- and trans- structures in PTMSP was calculated from the 13С NMR spectra using the Winnmr1d program (Bruker) suitable for calculation of incompletely resolved spectra. The ratio was calculated from the relative intensities of the peaks in the doublets. The peaks were assigned in [18] by combining the NMR and IR data with theoretical analysis of the normal modes of polymers with different microstructures.

(4) In Line 127, quaternization reaction was carried out at 60 deg C, while in Table 1, the reaction temp. is 55 deg C. Double check the reaction conditions.

Answer: Thank you, The correct temperature is 55°C, the tex is changed accordingly.

(5) Figure 2, the title of x-axis should be wavenumber.

Answer: Correction is done

(6) Delete Figure 3, because it duplicates the information already shown in Figure 1.

Answer: The Figure is deleted.

(7) Redo the XRD experiment, the baselines of the diffraction patterns are not even. Typically, the XRD baseline of PTMSP should be flat, as shown in doi.org/10.1016/j.seppur.2015.11.032, doi.org/10.1016/j.micromeso.2007.06.044, or doi.org/10.1016/S0969-806X(00)00218-8.

Answer: Thank you for the recommendation, the Figure is corrected.

(8) The baselines for the XRD patterns are so rough, so the peak half-width values might not be accurate.

Answer: The determination of the half-width was done after XRD diagrams were corrected for the baseline.

(9) In Table 4, present the uncertainty of gas transport properties, including permeability, diffusivity, and solubility.

Answer: The precision of the gas transport paremeters determination is now discussed in the text section related to the new Figure 6 showing the “time-lag” curves for CO2, CH4 and N2 for the case of PTMSP, containing 5 mol% butylimidazole bromide units.

(10) The authors have noted that quaternization significantly reduces CO2 (with the smallest kinetic diameter) diffusion coefficient, rendering it even smaller than that of N2, O2, and CH4 with the larger kinetic diameter and making CO2/N2 (or CH4) diffusion selectivity is less than 1. Why, what is the possible reason? Is that possible resulting from the uncertainty in determining the diffusivity coefficient?

Answer: We have added the Figure 6 presenting “time-lag” curves for PTMSP, containing 5 mol% butylimidazole bromide units. The following text was added as well:

At the same time it is necessary to note that CO2 at 30°C is sorbed in the synthesized polymers reversibly in contrast to e.g. quaternary ammonium compounds studied in [29] which can be judged from the time-lag and the corresponding diffusion coefficients (Figure 6). This is confirmed by the analysis of the “time-lag” curves for CO2, CH4 and N2. The “time-lag” curves of CO2 accumulation in the permeate volume of the experimental facility obtained one after another with a short, 2 min evacuation time between them show exactly the same behaviour as observed for other two gases presented on Figure 6. The time-lag deviation is very minor which results in just 1.2% error of CO2 diffusion coefficient fluctuation while the same value for CH4 is 6.3%. The experimental facility used for gas transport experiments has just 35 ms of systematic error of the time-lag determination and in the current study showed ±5% error in permeability coefficient detemination.

The behaviour of the polymer with ionic liquid is of a big interest but, unfortunately it is not in the scope of the current publication to carry out direct observations of solubility coefficients of CO2 in the synthesized polymers by gas sorption experiments. Authors are planning to study gas sorption, derive solubility coefficient and from permeability and solubility coefficient to derive the diffusion coefficient. Unfortunately it is not possible at this time.

The synthesized butylimidazole bromide containing polymers show fully reversible solubility of the CO2, which can be judged from exactly the same time-lag values for the CO2 determined at 30°C. This is in contrast to other quaternary ammonium compounds as e.g. in Shishatskiy et al. J. Membr. Sci. 2010, V. 359, P. 44 where it was clearly observed irreversible sorption of CO2 on active centers of quaternary ammonium compounds up to the temperature of 55°C. The origin of significant CO2 diffusion coefficient reduction in butylimidazole bromide containing polymers is not clear, it can be necessary to conduct NMR investigation of CO2 diffusion in these polymers and it’s interaction with functional groups of polymers.

(11) It’s good to show the time-lag plots for the determination of diffusivity coefficients.

Answer: Figure 6 is added demonstrating ime-lag curves for CO2, CH4 and N2 for the case of 5% butylimidazole bromide containing polymer.

(12) In Figure 6, put citations for the data points of ‘all polymers’ and ‘polyacetylenes’

Answer: The citation to the Membrane Society of Australasia is added.

(13) Misuse the quotations marks, for example, "Acros Organics" in Line 119, the "reflection" geometry in Line 162.

Answer: Corrected

Reviewer 3 Report

Please provide the detailed analysis and calculation for the degree of quaternization.   Include the elemental analysis data in the main text. Include the proton NMR spectra for 2.5mol% and 5mol% functionalized polymer in the main text. Figure 2 caption is wrong.  it should be IR spectra. Figure 6,   label the data points for quaternized PTMSP. (one for 2.5mol%, another for 5mol%)

Author Response

Reviewer 3

Comments and Suggestions for Authors

Please provide the detailed analysis and calculation for the degree of quaternization.

Answer: The following description is added:

Recalculation of the mass content of nitrogen in the molar content of quaternized units was carried out according to the formula:

Φ = xM3/(28 – x(M1 – M2 – М3))  (3),

where Φ is the molar fraction of N-containing units, x is the mass fraction of nitrogen in the sample (according to the results of elemental analysis), 28 is the relative atomic mass of nitrogen, M1, M2 and M3 are the molecular (molar) masses of the initial, brominated and quaternized elementary unit of PTMSP, respectively.

   Include the elemental analysis data in the main text.

Answer: done.

Include the proton NMR spectra for 2.5mol% and 5mol% functionalized polymer in the main text.

Answer: Unfortunately we can’t agree to include NMR study of synthesized polymers into the text of the manuscript. The synthesized functionalized polymers contain not just quaternized moieties but brominated and initial (or pristine) PTMSP as well, the interpretation of NMR spectra is of a great difficulty. It is generaly accepted, to our knowledge, to use element analysis and results of IR spectroscopy for the judgement on quaternized monomers content in the polymer. Please see e.g. the work of T. Masuda http://dx.doi.org/10.1016/j.polymer.2013.10.030

Figure 2 caption is wrong.  it should be IR spectra.

Answer: Corrected

Figure 6,   label the data points for quaternized PTMSP. (one for 2.5mol%, another for 5mol%)

Answer: The Figure (now Figure 7) is corrected, it contains points for initial PTMSp, brominated one and two quaternized polymers.

Round 2

Reviewer 2 Report

The authors well addressed the comments and the manuscript is publishable after minor revisions.

1) Figure 2 IR spectra, use 'Wavenumber' instead of 'ν' as the x-axis title. Remove labels of the y-axis (0,0 - 1,6).

2) Line 401, 'is shown on Figure 7.', should be Figure 6 instead.

Reviewer 3 Report

The authors has made improvement in the revised manuscript. It can be published as is.